# A potential diagnostic serum immunological marker panel to differentiate between primary and secondary knee osteoarthritis

**Sakuni Rankothgedera**[1], **Inoshi Atukorala**[2], **Chandrika Fernando**[3], **Duminda Munidasa**[4], **Lalith Wijayaratne**[5], **Preethi Udagama**[1]*

**1** Faculty of Science, Department of Zoology & Environment Sciences, University of Colombo, Colombo, Sri Lanka, **2** Faculty of Medicine, Department of Clinical Medicine, University of Colombo, Colombo, Sri Lanka, **3** Faculty of Science and Engineering, School of Electrical Engineering, Computer and Mathematical Sciences (EECMS), Curtin University, Perth, Western Australia, **4** Rheumatology & Rehabilitation Hospital, Ragama, Sri Lanka, **5** Nawaloka Hospital PLC, Colombo, Sri Lanka

* preethi@zoology.cmb.ac.lk

**Data Availability Statement:** The demographic and clinical data used to support the findings of this study are restricted by the Ethics Review Committee of Faculty of Medicine in University of

## Abstract

Inflammation contributes to knee osteoarthritis (KOA) where many immunological mediators participate in its initiation and progression. Most clinicians manage primary (pKOA) and secondary osteoarthritis (sKOA) alike. Whether immunological profiles of pKOA and sKOA differ remains obscure. Hence, we aimed to differentially identify potential serum immunologic diagnostic markers of pKOA and of sKOA. This case control study used 46 KOA patients (pKOA, n = 30; sKOA, n = 16), and 60 age, gender matched controls (normal healthy, n = 30; systemic lupus erythematosus [SLE] disease controls, n = 30) where serum was assayed for cytokines (TNF-α, IL-1β, IL-6, IL-10) and nitric oxide derivatives (NOx). Sandwich ELISA assessed cytokine levels, while the 'Griess assay' quantified NOx levels. The diagnostic accuracy of optimal marker combinations was evaluated by the CombiROC web tool. Compared with pKOA, sKOA serum displayed significantly elevated levels of pro inflammatory cytokines (TNF-α, IL-1β, IL-6) with a concurrent decrease in the anti-inflammatory cytokine, IL-10 (P<0.05). This was reiterated by significantly higher Th1:Th2 (TNF-α: IL-10) serum cytokine ratio observed in sKOA compared to that of pKOA. The CombiROC curves identified TNF-α, IL-1β, IL-6 and NOx as the best performing panel of potential diagnostic markers to discriminate pKOA from control groups (~97% accuracy, 90% Sensitivity [SE] and 98% specificity [SP]), while TNF-α, IL-1β and IL-6 discriminated sKOA from control groups (~100% accuracy, 100% SE, and 98% SP). The study identified discrete serum immune biomarker panels to differentiate between pKOA (TNF-α, IL-1β, IL-6 and NOx) and sKOA (TNF-α, IL-1β and IL-6). These findings may assist in developing distinct therapeutic agents for the two types of KOA.

Colombo, Sri Lanka (Reference No: EC/17/053)in order to protect privacy of volunteers. Data are available from Preethi V. Udagama, preethi@zoology.cmb.ac.lk (University of Colombo) for researchers who meet the criteria for access to confidential data. We are unable to give a non author contact for request of data, as per the above statement and this ethics committee will not take responsibility to provide our data on request. The only alternative would be to provide the first author's current contact which is svrankot@central.uh.edu. Although the authors cannot make their study's data publicly available at the time of publication, all authors commit to make the data underlying the findings described in this study fully available without restriction to those who request the data, in compliance with the PLOS Data Availability policy. For data sets involving personally identifiable information or other sensitive data, data sharing is contingent on the data being handled appropriately by the data requester and in accordance with all applicable local requirements. Data are available from all authors (email addresses available on first page of publication) for requests made via email.

**Funding:** Provision of special funds to PU by the Vice Chancellor, University of Colombo for interdisciplinary undergraduate research carried out between the Science and Medical Faculties of the University of Colombo, Sri Lanka. The funder had no role in study design, data collection and analysis, decision to publish, or preparation of the manuscript.

**Competing interests:** The authors have declared that no competing interests exist.

## Introduction

Osteoarthritis (OA) encompasses a heterogenic group of joint diseases [1] and was popularly known as "non-inflammatory arthritis" until studies conducted during the past few years changed this ideology [2]. Investigations revealed that low-grade, chronic inflammation plays a key role in the pathophysiology of OA [2]. In recent years, many studies concentrated on cellular inflammation and production of inflammatory mediators associated with synovitis [3]. A variety of inflammatory mediators including cytokines and nitric oxide derivatives (NOx) are produced by multiple joint tissues and cells and promote inflammatory and catabolic processes in chondrocytes [4,5]. Many of these mediators are biomarkers of inflammation and have potential to be useful in developing novel therapeutic options based on the current understanding of molecular pathogenesis of OA [6]. Therefore, many investigators focus on assessing the clinical utility of these inflammatory mediators in the early diagnosis and prognosis of the disease [7].

Knee OA (KOA) is one of the most common types of OA and is defined by degeneration of the knee's articular cartilage [8]. It is established that pro inflammatory cytokines, particularly IL-1β and TNF-α, play a crucial role in the initiation and development KOA. IL-1β is responsible for cartilage destruction, while TNF-α drives the inflammatory process. Both these mediators can induce chondrocytes and synovial cells to produce other pro inflammatory cytokines such as IL-6 [4,9]. IL-6, being yet another pro-inflammatory cytokine, increases the number of inflammatory cells in synovial tissue, stimulating the proliferation of chondrocytes and amplifying the effects of IL-1β [9] whereas IL-10 is an anti-inflammatory cytokine which reduces inflammation by down-regulating the pro inflammatory cytokines [4]. Further, NOx are also involved in inflammatory processes in KOA [5]. NOx contributes to the cartilage destruction in KOA by enhancing expression of matrix metallo-proteinases (MMPs), inhibition of synthesis of collagen and proteoglycan and inducing apoptosis of chondrocytes [10].

The American College of Rheumatology (ACR) classifies KOA into two groups; primary (pKOA) and secondary (sKOA) [11]. Conditions such as trauma, deformities in bones and joints, obesity, bone and joint disorders such as avascular necrosis, rheumatoid arthritis (RA), osteoporosis, and endocrine diseases and calcium deposition disease can be used to distinguish sKOA from pKOA, as the former lacks clear etiopathogenesis [11,12].

Studies confirm the incidence of many different types of immune cells in KOA synovium, and amongst these are T helper cells which are classified as Th1 and Th2 cells by their pattern of cytokine production. A Th1:Th2-type cytokine imbalance with Th1-type cytokine predominance is believed to be significant in RA [13,14]. With the emergence of evidence for the role of inflammation in [13,15], we propose that Th1:Th2 cytokine ratio can be used to further explore the inflammation manifested in pKOA and sKOA.

The distinction between pKOA and sKOA is not taken into consideration by clinicians during management of patients [12]. Therefore, this study aimed to evaluate the relationship of selected immunologic mediators, i.e. cytokines (TNF-α, IL-6, IL-1β and IL-10) and NOx, with inflammation and tissue damage in cohorts of pKOA and sKOA patients in Sri Lanka.

## Materials and methods

### Study design

This case control study consisted of patients with pKOA and sKOA as the "cases", patients with SLE as the "disease control" group and normal, healthy individuals as the "control" group. The study was conducted in accordance with ethical considerations of the Ethics Review Committee of the Faculty of Medicine, University of Colombo, Sri Lanka (Reference

No: EC/17/053) and the principles laid down in the Declaration of Helsinki and its later revisions.

## Sample size

The sample size for the study was determined using Fisher's formula for sample size determination; $n = Z^2Pq/d^2$, where n = Desired sample size population <10,000, Z = Standard normal deviation (set at 1.96 at 95% confidence level), P = Proportion of the subjects (estimated global age-standardized prevalence of KOA = 3.8%), q = 1 and P = absolute precision or sampling error tolerated (set at 5%) [16–18]. When the formula was applied to the worldwide prevalence of OA, the sample size (n) obtained was 56.

Sample size of n = 30 per group was selected as the most adequate sample, due to logistic difficulties and reagent costs involved, which would not compromise the accuracy of the statistical analysis. Hence, a non-parametric approach was adopted to carry out the statistical analyses [19].

## Study participants

Fig 1 presents the study scheme which elucidates the participants of this case control study, recruited under four groups (two case and two control groups), with inclusion and exclusion criteria. Serum samples were obtained from 30 normal, healthy volunteers attached to the staff of the University of Colombo for the study group of normal healthy controls. During recruitment, individuals with malignant tumors, diabetes, hypertension, high blood cholesterol, liver, kidney or heart conditions, any systemic condition, recent bacterial or viral infection that may cause an elevation in inflammatory mediators or individuals under any type of medication were excluded.

The disease control group comprised patients diagnosed with SLE, who were specifically treated for lupus arthritis. The SLE patients attending the University Clinic of the National Hospital Sri Lanka (NHSL) were recruited as disease controls.

A group of 30 individuals diagnosed with Stage 3 pKOA based on clinic, laboratory and radiolergical findings pertaining to criteria defined by the American College of Rheumatology (ACR) (Mean duration of condition- 6.93 ± 3.78 years), attending the clinic at Rheumatology and Rehabilitation Hospital, Ragama, Sri Lanka were recruited to the study.

Stage 3 sKOA patients (n = 16), a heterogenic group of patients (Mean duration of condition—6.44± 3.68 years), were recruited based on the scheme of classification established by the ACR and with a history of knee joint injuries, avascular necrosis, scoliosis or any other joint deformity leading to KOA. This group was recruited from both the Rheumatology and Rehabilitation Hospital, Ragama, and the Orthopedic Clinic of the NHSL.

The test subjects (pKOA and sKOA) and the control subjects (normal healthy and SLE) recruited to the study were matched for age and gender.

## Collection of patient data and blood samples

Socioeconomic data and information related to disease history were obtained by an interviewer administered questionnaire. All test subjects completed the KOOS (Knee injury and Osteoarthritis Outcome Score) questionnaire (Version LK 1.0 –English) [20].

Blood (5.0 ml) was obtained from each subject after voluntary informed consent. Blood was allowed to clot in a plain blood collection tube at room temperature. The clot was separated by centrifuging at 1000 x g for 15minutes in a refrigerated centrifuge, and the supernatant (serum) was collected, apportioned into 0.5 ml aliquots and stored at –20˚C until further use [21].

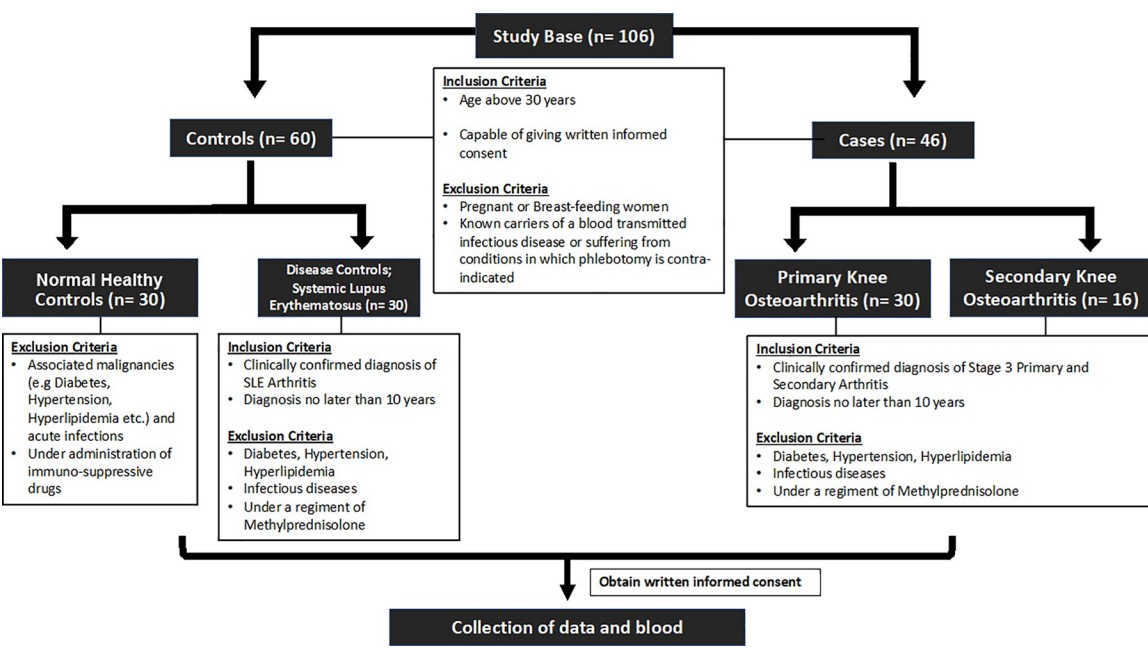

**Fig 1. Scheme of the case control study design.**

## Assay of serum analytes

The selected serum cytokine levels (TNF-α, IL-1 β, IL-10 and IL-6) of the subjects were assayed using human sandwich ELISA kits, according to the specifications of the manufacturer (Human BD OptEIA$^{TM}$Set- BD Biosciences, USA). Serum NOx levels were determined by performing the Griess assay according to standard procedure [22], where 50–70μl of zinc sulfate (15 mg/mL) was added to 200μl of the serum sample and centrifuged at 1000 x g for 10 minutes for the de-proteinization of the sample and a portion of 100 μl of the supernatant was transferred to a micro well plate. 100 μl vanadium (III) chloride was dispensed into each well followed by 100 μl of Griess reagent for nitrite (Sigma Aldrich 03553). Samples were incubated for 30 minutes at 37˚C. The absorbance was measured at 540 nm. Serum NOx concentration was established from the linear standard curve plotted by 0–100 μM sodium nitrate [23].

## Statistical analyses

SPSS software, version 21.0 (IBM, USA) was used for statistical analyses. The Chi-square, Mann-Whitney U and Kruskal–Wallis tests were used to compare the demographic data and serum levels of immunological mediators among subject groups. Results were presented as mean ± SD. Correlations between analytes as well as among analytes and KOOS score were calculated using Spearman's correlation coefficient. Binary logistic regression was used to assess the risk imposed by various confounding factors (sex, consumption of dairy food, having a history of bone related surgeries and a family history of rheumatic diseases) on pKOA and sKOA over controls. CombiROC was used to plot combined ROC curves to determine the best suited combinations of biomarkers with corresponding sensitivity, specificity, and accuracy (area under the curve [AUC]). The level of statistical significance was set at $p \leq 0.05$ [24].

**Table 1. General characteristics of the recruited control and case groups.**

| | Control groups | | Case groups | | p value# |
|---|---|---|---|---|---|
| | Normal healthy (n = 30) | Disease Control (n = 30) | pKOA (n = 30) | sKOA (n = 16) | |
| Mean Age (years) | 56.8± 5.242 | 54.23± 5.380 | 55.80± 3.428 | 56.81± 7.441 | 0.251 |
| Gender (% of female) | 96.67 | 96.67 | 96.67 | 81.25 | 0.112 |
| BMI | 23.36±3.55 | 22.33±6.07 | 22.94±2.39 | 23.62±5.45 | 0.735 |

pKOA–primary knee osteoarthritis; sKOA–secondary knee osteoarthritis.

#p value—for the comparison between control and case groups (Kruskal-Wallis H test).

## Results and discussion

Table 1 presents the general characteristics of the recruited study groups. The case groups and the control groups displayed no significant differences in mean age, gender and BMI at a 95% confidence interval ($p > 0.05$).

### Serum concentration of analytes

When compared to normal healthy controls the pKOA test group showed a significant elevation in the serum levels of the four analytes, TNF-α, IL-10, IL-1β ($p < 0.0001$) and NOx ($p = 0.004$). The sKOA group displayed significant elevation of serum levels of all five analytes TNF-α, IL-10, IL-6 ($p < 0.0001$), IL-1β ($p = 0.007$) and NOx ($p = 0.041$) in comparison to normal healthy controls (Fig 2A and 2B, Table 2).

pKOA patients showed an approximate two-fold increase in the TNF-α concentration ($p < 0.0001$) over disease controls, whereas the rise in the serum concentration of IL-1β was slight yet statistically significant ($p < 0.0001$). When compared with disease controls, serum IL-

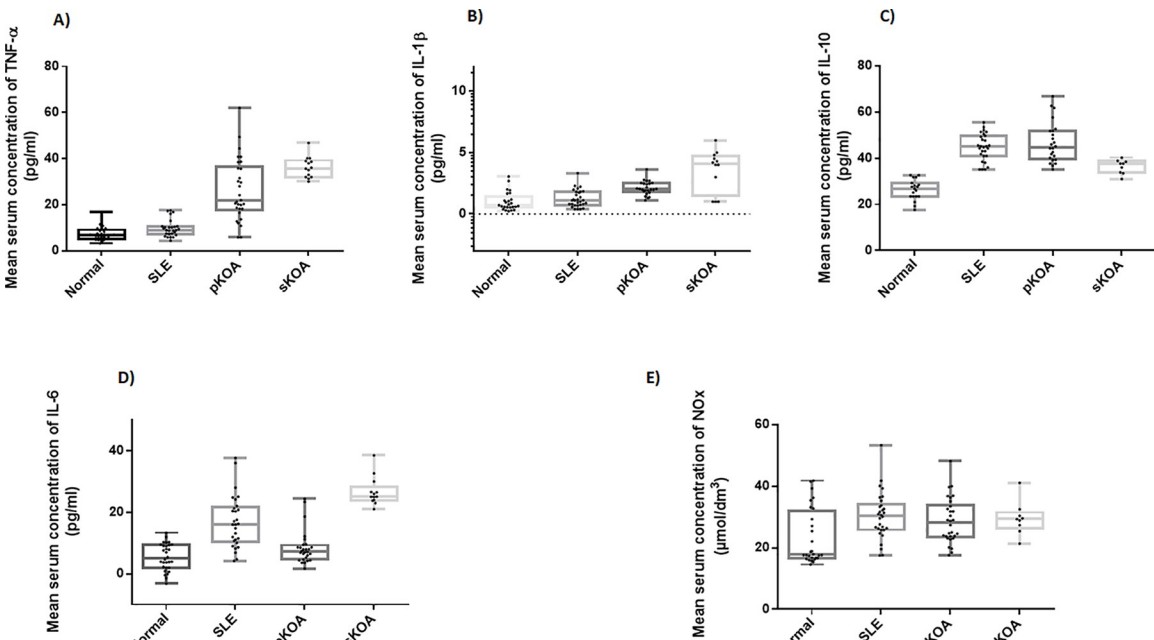

**Fig 2. Serum concentrations of the Analytes, Cytokines (A)TNF-α, (B) IL-1β, (C) IL-10, (D) IL-6 and (E) nitric oxide derivatives (NOx) (Whiskers extend from the smallest value to the largest value of each data set; Normal: Normal, healthy controls, SLE: Systemic Lupus Erythematosus disease controls, pKOA: Primary knee osteoarthritis, sKOA: Secondary knee osteoarthritis).**

**Table 2. Mean serum levels of the analytes assayed in case and control groups.**

| | Control groups (n = 60) | | | | Case groups (n = 46) | | | |
| --- | --- | --- | --- | --- | --- | --- | --- | --- |
| | Normal healthy (n = 30) | | Disease Control (n = 30) | | pKOA (n = 30) | | sKOA (n = 16) | |
| | Mean conc. | 95% CI for mean (lower limit–upper limit) | Mean conc. | 95% CI for mean (lower limit–upper limit) | Mean conc. | 95% CI for mean (lower limit–upper limit) | Mean conc. | 95% CI for mean (lower limit–upper limit) |
| TNF-α (pg/ml) | 9.76 ±7.92 | 6.79–12.72 | 11.27±5.88 | 9.08–13.47 | 26.24 ±13.38 | 21.24–31.2 | 36.07 ±5.6 | 32.97–39.2 |
| IL-6 (pg/ml) | 6.56 ±3.63 | 5.10–8.03 | 18.92 ±7.75 | 15.65–22.19 | 9.2 ±6.56 | 6.71–11.7 | 22.24 ±19.13 | 12.05–32.43 |
| IL-1β (pg/ml) | 1.13 ±0.82 | 0.73–1.53 | 1.25 ±0.69 | 0.98–1.51 | 3.45 ±4.67 | 1.64–5.26 | 4.12 ±3.92 | 1.95–6.29 |
| IL-10 (pg/ml) | 27.25 ±5.32 | 25.14–29.36 | 45.58 ± 9.94 | 41.87–49.29 | 42.47 ±12.52 | 37.7–47.23 | 38.39 ±8.83 | 33.68–43.1 |
| NOx (μmol/dm$^3$) | 23.29 ±9.02 | 19.8–26.79 | 30.3 ±7.61 | 27.46–33.14 | 28.74 ±7.29 | 26.01–1.46 | 26.46 ±5.68 | 23.32–29.61 |

pKOA–primary knee osteoarthritis; sKOA–secondary knee osteoarthritis.

6 level was significantly lower ($p<0.0001$) in pKOA patients. Serum levels of both TNF-α ($p<0.0001$) and IL-1β ($p = 0.011$) of sKOA patients were approximately three folds higher of that of the SLE patients, while IL-10 ($p = 0.005$) and NOx ($p = 0.05$) were significantly lower in sKOA patients than in the disease controls. IL-6 levels of sKOA subjects were on par with levels of SLE subjects ($p = 0.868$) (Fig 2A and 2B, Table 2).

Considering the two case groups, TNF-α ($p = 0.006$) and IL-6 ($p = 0.001$) levels showed a steep increase from pKOA to sKOA. Serum IL-10 levels of sKOA was significantly lower than that of the pKOA group ($p = 0.05$). Both serum IL-1β and NOx levels showed no discrepancy in the two KOA groups ($p>0.05$) (Fig 2A and 2B, Table 2).

## Th1:Th2 cytokine ratio

Among all study groups the highest TNFα: IL-10 (Th1: Th2) cytokine ratio was observed in patients clinically diagnosed with sKOA while the lowest was in the disease controls. Among the two test groups, sKOA showed a cytokine ratio that was significantly Th1 skewed ($p<0.0001$) than that of pKOA patients (Fig 3). The cytokine ratio of TNF-α: IL-10 of the normal healthy control group was significantly lower than that of the test populations ($P<0.0001$). Comparison with the disease controls revealed that both test groups showed significantly elevated cytokine ratios ($p<0.0001$ for both OA types) (Fig 3). However, the assay of additional serum cytokines such as IFN-γ, IL-2, IL-12 Vs IL-4, IL-5 would have explicitly defined the Th1:Th2 balance among the study groups. This was not pragmatic due to restricted funds.

## Association of Th1: Th2 cytokine ratio and the KOOS pain score with serum levels of analytes

A bivariate correlation was carried out to investigate the association of the Th1: Th2 cytokine ratio and the KOOS pain sub score of pKOA and sKOA patients obtained through the KOOS questionnaire, with serum levels of cytokines IL-6 and IL-1β, and NOx.

While a significant correlation of Th1:Th2 cytokine ratio with IL-1β concentration ($p = 0.01$) was evident, a significant negative correlation was established with the KOOS pain score ($p<0.001$). No significant correlations were established with KOOS pain score and the tested serum concentrations of analytes ($p>0.05$).

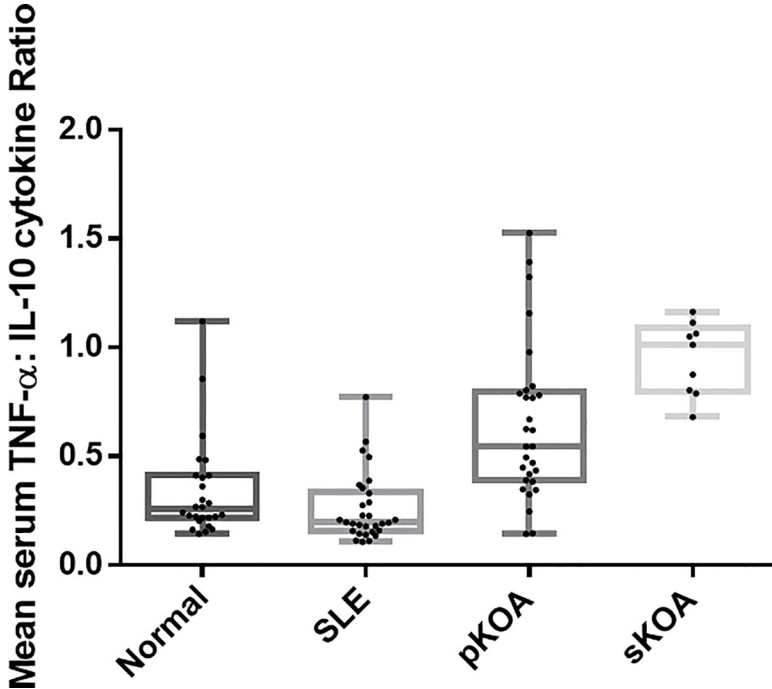

**Fig 3. TNF-α: IL-10 (Th1: Th2) cytokine ratio of the four study groups (Whiskers extend from the smallest value to the largest value of each data set; Normal: Normal, healthy controls, SLE: Systemic Lupus Erythematosus disease controls, pKOA: Primary knee osteoarthritis, sKOA: Secondary knee osteoarthritis).**

## CombiROC analysis

The optimal marker combinations were investigated using combiROC tool available at http://CombiROC.eu which is designed to select multi-marker signatures from a panel of markers analyzed (Table 3). The best individual and combination of markers (Gold markers & combos) are selected based on the highest area under the curve (AUC), sensitivity (SE) and specificity (SP). Fig 4A and 4B and Table 3 present the obtained gold markers on multiple Receiver Operating Characteristic (ROC) curves.

When uploading the data of the analytes into the tool, a test cut off value of 34.64 (control mean + SD) was used since this was the cut off value at which reliable outcomes with high SE

**Table 3. The performance of individual analytes as gold markers or combination of markers via Receiver Operating Characteristic (ROC) curve analysis.**

| Test Group | Markers | AUC | SE% | SP% | Optimum cut-off |
|---|---|---|---|---|---|
| pKOA | TNF-α &NOx (Combo I) | 0.881 | 0.933 | 0.750 | 0.233 |
| | TNF-α, IL-6 &NOx (Combo II) | 0.912 | 0.933 | 0.833 | 0.335 |
| | TNF-α, IL-1β &NOx (Combo III) | 0.956 | 0.933 | 0.867 | 0.243 |
| | TNF-α, IL-6, IL-1β &NOx (Combo IV) | 0.973 | 0.900 | 0.983 | 0.582 |
| sKOA | TNF-α (Marker 1) | 0.982 | 1 | 0.933 | 0.207 |
| | TNF-α & IL-6 (Combo I) | 0.993 | 1 | 0.950 | 0.206 |
| | TNF-α & IL-1β (Combo II) | 0.985 | 1 | 0.950 | 0.351 |
| | TNF-α, IL-6 & IL-1β (Combo III) | 0.996 | 1 | 0.983 | 0.462 |

pKOA–primary knee osteoarthritis; sKOA–secondary knee osteoarthritis.

AUC—Area under the curve; SE–Sensitivity; SP–Specificity.

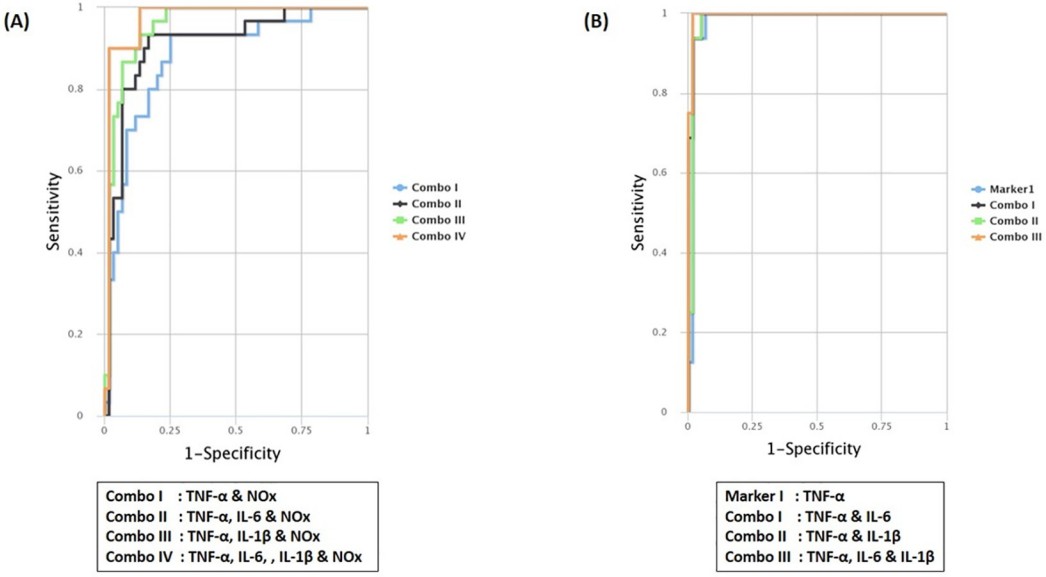

**Fig 4. Multiple Receiver Operating Characteristic (ROC) curves as a function of sensitivity vs 1-specificity to compare the performance of markers and combos for (A) Primary Knee Osteoarthritis (pKOA), (B) Secondary Knee Osteoarthritis (sKOA).**

and SP were obtained. Among the gold combinations, TNF-α performed well as a potential diagnostic marker for both test groups by being included in all the gold combinations and also as a gold marker for sKOA. Interestingly, IL-10 was not included in any of the gold combinations for either of the two test groups.

It was evident that NOx combined with other markers can be a potential marker panel in the diagnosis of pKOA and therefore highly useful as a discriminative marker between the two groups of KOA.

CombiROC performs a 10 fold cross validation (10 CV) to test the reliability of the panel of analytes as a valid marker panel. Yet to overcome the limitation of over-optimistic results of CV, a permutation test is carried out.

The plots obtained via the permutation test are shown in Fig 5A and 5B while Table 4 presents the accuracy (ACC) and error rate of the 10 CV model and the permutated models, as well as the SE, SP and AUC. The plots present the density plot of the AUC values for the ROC curves of the panel of markers in the analysis of 500 permutation test and the grey line represents the real AUC value. Our results suggest that the real AUCs are found outside the normal distribution of most likely AUCs.

## Significance of non-confounding and confounding factors for primary and secondary KOA

The risk factors for KOA taken into account were age, sex, BMI, frequency of consumption of animal proteins (dairy based products, eggs, fish, meat), usage of protein supplements, diabetes mellitus, hypertension, high blood cholesterol, asthma, hysterectomy, surgeries and accidents affecting bones and the family history of rheumatic diseases.

Of these, consumption of eggs, meat and fish appeared to be a significant non-confounding factor for both pKOA and sKOA ($p = 0.002$). Having a history of hysterectomy seemed a significant factor for both types of KOA ($p = 0.004$). Female sex ($p = 0.023$), current state of diabetes ($p = 0.003$) and hypertension ($p = 0.007$) and family history of rheumatic diseases ($p =$

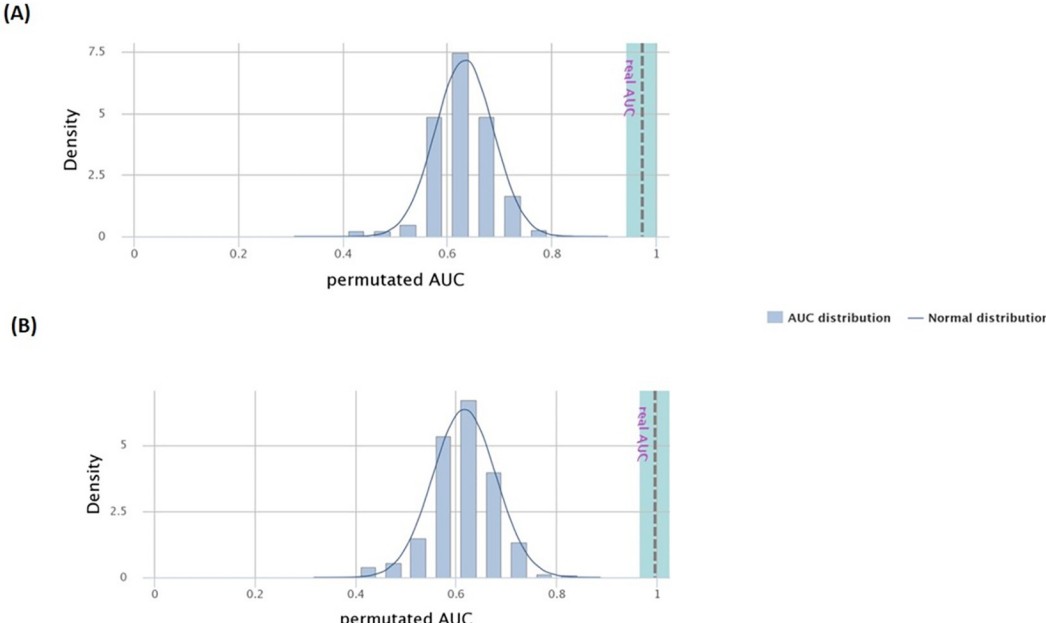

**Fig 5. Distribution of the permutated Area under the curves (AUCs) in comparison to normal distribution and the significance of the real AUC for the best performed marker panel of (A) Primary Knee Osteoarthritis (pKOA), (B) Secondary Knee Osteoarthritis (sKOA).**

0.003) were found to be significant non confounding factors for pKOA while having a history of bone related accidents ($p = 0.001$) or surgeries ($p<0.001$) were exclusively significant for sKOA. The significant non-confounding factors were fitted into a binary logistic regression model to investigate the confounding effect held by the factors over pKOA and sKOA. The developed model established that daily consumption of fish and meat may reduce the risk of pKOA over normal healthy individuals by 0.92 folds ($p = 0.014$ and $p = 0.025$, respectively; CI at 95%: 0.011–0.607 and 0.009–0.730, respectively). A history of bone related accidents may increase the risk of sKOA over a normal healthy individual by 32 folds ($p = 0.037$; CI at 95%: 1.237–822.672).

Confounding factors incorporated into the binary regression model but not deemed statistically significant were the sex, consumption of dairy food, having a history of bone related surgeries and a family history of rheumatic diseases ($p = 0.597, 0.142, 0.157$ and $0.1$, respectively).

Our results suggest that selected serum immunological mediators such as TNF-α, IL-6, IL-1β, IL-10 and NOx may be used as potential diagnostic biomarkers for the differential detection of pKOA and sKOA.

**Table 4. Comparison of the performance of the 10 CV (10 fold cross validation) model and the permutated model.**

|  |  | ACC | Error rate | SE | SP | AUC |
|---|---|---|---|---|---|---|
| pKOA | 10 fold CV | 0.922 | 0.078 | 0.900 | 0.933 | 0.924 |
|  | Permutated model | 0.639 | 7.956 | 0.633 | 0.659 | 0.632 |
| sKOA | 10 fold CV | 0.947 | 0.053 | 1.00 | 0.933 | 0.978 |
|  | Permutated model | 0.636 | 6.824 | 0.63 | 0.654 | 0.616 |

pKOA–primary knee osteoarthritis; sKOA–secondary knee osteoarthritis.

ACC–Accuracy; SE–Sensitivity; SP–Specificity; AUC—Area under the curve.

The use of a disease control group alongside the healthy control group in the current case–control study suggested the unique identification of pathophysiology that may occur exclusively in pKOA and sKOA in contrast to SLE. Arthritis is one of the most common signs of SLE [24] that is often present at the time of diagnosis of SLE patients. Yet, SLE is a chronic inflammatory autoimmune disease alike RA and therefore unlike in KOA, auto antibodies play a major role in lupus arthritis which is similar to the pathogenesis of RA. As such, immunological markers can be earmarked to diagnose pKOA and sKOA independent of arthritis conditions caused by auto antibodies. The variations of the levels of the selected serum immunologic mediators analyzed in KOA patients compared with disease controls implicate the ability of these analytes to be used in understanding the diagnosis of numerous rheumatologic diseases extensively. Nonetheless, the low sample size of the sKOA group may present as a potential limitation of this study, which may perhaps have ramifications.

In the current study, when compared with normal healthy controls, all serum analytes showed an increase in concentration in both pKOA and sKOA. This observation is in agreement with previous studies on pKOA [11,25–27] nevertheless no literature was found for the serum cytokine profile of sKOA. This elevation of both pro and anti- inflammatory cytokines and NOx in KOA patients over healthy controls may be attributed to the inflammation driven metabolic processes in KOA patients absent in normal controls.

In comparison to the disease controls (SLE patients), significantly higher serum levels of all pro inflammatory cytokines tested were detected in both pKOA and sKOA patients. Being an autoimmune condition, the reason for disease controls to manifest low levels of the pro inflammatory cytokines TNF-α and IL-1 β may be, that the SLE population recruited to the study were under medication at the time of recruitment and no active inflammation was therefore observed. Conversely, the proinflammatory cytokine IL-6 showed a significant rise in SLE patients exhibiting the major contribution of IL-6 to the disease pathology of SLE acting as the key cytokine alongside IFN-γ [27]. IL-10 which shows an antagonistic effect to pro-inflammatory cytokines such as TNF-α and IL-1β was low in both types of OA further accounting for the higher status of inflammation in KOA patients than in the SLE patients enlisted in this study.

The significant elevation of pro-inflammatory cytokines and the concurrent drop in the anti-inflammatory cytokine, IL-10 in serum of sKOA patients than in pKOA patients suggests that inflammation in the former may be more significant than in the latter. Though serum levels of pro-inflammatory cytokines suggested that sKOA patients may manifest higher inflammatory responses, the converse was observed in their serum NOx levels. Recent evidence suggests contradictory roles of NOx in OA where although NOx is associated with inflammation in OA, nitric oxide (NO) and its redox derivatives may also play protective roles in the joint. A catabolic role of NO which leads to the development of OA and its inflammatory response was reported. There is also evidence that adding NO into cultured chondrocytes externally may interfere with the Nuclear Factor-κB (NF-κB) pathway and play an anti-inflammatory role while stimulating collagen synthesis [26].

A Th1:Th2 cytokine imbalance with Th1-type cytokines predominating has been suggested to be of pathogenetic significance in RA; High levels of Th1-type cytokines play a crucial role in locinflammation in RA [28]. With the emergence of evidence for the role of inflammation in KOA, the Th1:Th2 cytokine ratio can be used to further explore the inflammation present in pKOA and sKOA [29]. This was further reinforced in the current study by the significant positive correlation between Th1:Th2 cytokine ratio and IL-1β, a pro-inflammatory cytokine. Yet, the absence of an elaborate Th1/Th2 characterization using more cytokines is one of the limitations of the current study.

The cause of 60% of sKOA cases was physical trauma, where they were suffering from OA for a period of 1 to 2 years. Among the other causes were, avascular necrosis, scoliosis and apatite arthropathy whereas, the cause of pKOA subjects recruited was aging and "wear and tear" of joints associated with aging. The vital role of inflammation in sKOA over pKOA may be due to the high concentration of damage associated molecular patterns (DAMPs) involved with sKOA patients.

The Knee injury and Osteoarthritis Outcome Score (KOOS) is an extension of the WOMAC Osteoarthritis Index to evaluate short-term and long-term symptoms and function in subjects with knee injury and osteoarthritis. Usually, the KOOS score ranges from 0 to 100 percent, with 0 representing a highly severe form of the disease while 100 represents the absence of the disease condition [20]. KOOS comprises of 5 subscales: Pain, other Symptoms, Function in daily living (ADL), Function in sport and recreation (Sport/Rec) and knee related Quality of life (QOL). The pro inflammatory cytokines, IL-1β and IL-6 showed negative correlation with the KOOS score depicting that inflammation is positively correlated with pain whereas the anti-inflammatory cytokine IL-10 positively correlated with the pain score and hence inversely correlated with pain indicative of the effect of the anti-inflammatory cytokine antagonistic to the pro-inflammatory cytokines. That pain positively correlated with the Th1: Th2 cytokine ratio, further confirmed that pain is indeed one of the classic symptoms of inflammation; This is mainly due to the sensitization of fine un-myelinated sensory nerves present in the osteoarthritic joint [30].

We used CombiROC, which is still a novel technique to identify all the possible and best performing markers and combinations of markers and to assess the validity of them as potential diagnostic biomarkers. There were no individual markers identified as "Gold" markers for pKOA [24]. Nevertheless, several combos were identified to be of high potential to become diagnostic markers of pKOA. Of those, the combination with the highest AUC (0.973), SE (0.900) and SP (0.983) was the combination of pro inflammatory cytokines, TNF-α, IL-6 and IL-1β and the inflammatory mediator, NOx. Therefore, it can be deduced that even though NOx did not show a significant difference among pKOA and sKOA groups, combined with other markers it too can be a potential diagnostic marker. TNF-α appeared to be an individual "gold" marker of sKOA, with a high AUC of 0.982, SE of 1 and SP of 0.933. Of the best performed combinations, TNF-α, IL-1β and IL-6 show the highest values for AUC (0.996), SE (1) and SP (0.983). TNF-α was observed to be common to both types of OA as it was included in the "gold" combinations of bio markers in both types of OA. Importantly, NOx appeared to be explicit for the detection of pKOA.

In our study, the demographic factors which have been recognized as risk factors of KOA by previous studies were taken into consideration [31–33] and the outcomes of this study agree with these previously established findings. The female sex which was found to be a significant non confounding factor is believed to be a greater risk factor of OA than the male sex due to multifactorial reasons such as the differences in knee anatomy, kinematics, genetics and hormonal influences between males and females [34]. According to ACR criteria, trauma is one of the three major causes of sKOA and our findings exhibit that traumatic surgeries and accidents related to bones are significant non confounding factors for sKOA whereas bone related accidents proved to be a significant confounding factor for sKOA as well. Our findings suggest that hysterectomy, a surgical procedure that involves the removal of all or part of the uterus (in some cases, along with the ovaries) is a significant non confounding factor of both types of KOA. Hysterectomies cause reduction in the production of the female sex hormone, estrogen. Agreeing with the results obtained, there is increasing evidence that estrogen fulfil a role in maintaining the homeostasis of articular tissues and, hence, of the joint itself. Though it appears that estrogen may have a beneficial effect on cartilage, the exact mechanism of this

protection has yet to be elucidated. Changes in the production of other hormones and growth factors caused by the hysterectomy may also affect osteoarthritis in women [35]. Scientific studies have expounded that inclusion of specific foods in the diet can strengthen bones, muscles, and joints and help the body to fight inflammation and disease. Consumption of animal proteins (milk, eggs, fish, and meat) was found to be a confounding as well as a non-confounding factor that affected both pKOA and sKOA. References can be found to illustrate that these animal proteins not only strengthen bones but products such as fish oil could also act as disease-modifying anti-rheumatic drugs (DMARDs) which is a category of drugs used in rheumatic conditions to slow down disease progression [36–38].

This study importantly highlights the potential of using selected biomarkers in the differential diagnosis of pKOA from sKOA. However, since no previous studies were reported on the performance of the assayed analytes in disease diagnosis of sKOA, except for their augmented serum levels in pKOA, these findings require further validation with significantly larger numbers of subjects of similar cohorts. This analysis may be further improved by incorporating other serum bio markers which are included in the panel of "12 OA related Biomarkers" by the FNIH OA Biomarkers Consortium Project [39]. Inclusion of the comprehensive details of the regimes of medication the recruits are prescribed with at the time of data collection would furthermore improve the final results of the study.

## Conclusion

In summary, this prototype study suggests that significantly higher levels of inflammatory serum cytokines are present in sKOA compared to pKOA. Furthermore, serum immune biomarker panels to differentiate between pKOA and sKOA were identified, which may assist in developing distinct therapeutic agents for the two types of KOA.

## Acknowledgments

Dr. V. Swarnakumar (National Hospital of Sri Lanka, Colombo 07, Sri Lanka) for assisting in sample collection is acknowledged.

## Author Contributions

**Conceptualization:** Sakuni Rankothgedera, Preethi Udagama.

**Formal analysis:** Sakuni Rankothgedera, Chandrika Fernando.

**Funding acquisition:** Preethi Udagama.

**Investigation:** Sakuni Rankothgedera, Inoshi Atukorala, Lalith Wijayaratne.

**Methodology:** Chandrika Fernando, Preethi Udagama.

**Resources:** Inoshi Atukorala, Duminda Munidasa, Lalith Wijayaratne.

**Supervision:** Lalith Wijayaratne, Preethi Udagama.

**Validation:** Chandrika Fernando, Duminda Munidasa.

**Visualization:** Sakuni Rankothgedera.

**Writing – original draft:** Sakuni Rankothgedera.

**Writing – review & editing:** Inoshi Atukorala, Preethi Udagama.

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
