## [Decision Letter · Decision Letter 0]

6 Jun 2021

PONE-D-21-12809

A potential diagnostic serum immunological marker panel to differentiate between primary and secondary knee osteoarthritis

PLOS ONE

Dear Dr. Udagama,

Thank you for submitting your manuscript to PLOS ONE. After careful consideration, we feel that it has merit but does not fully meet PLOS ONE’s publication criteria as it currently stands. Therefore, we invite you to submit a revised version of the manuscript that addresses the points raised during the review process.

We look forward to receiving your revised manuscript.

Kind regards,

Oreste Gualillo, PharmD, PhD

Academic Editor

PLOS ONE

Journal Requirements:

2a) If there are ethical or legal restrictions on sharing a de-identified data set, please explain them in detail (e.g., data contain potentially identifying or sensitive patient information) and who has imposed them (e.g., an ethics committee). Please also provide contact information for a data access committee, ethics committee, or other institutional body to which data requests may be sent.

2b) If there are no restrictions, please upload the minimal anonymized data set necessary to replicate your study findings as either Supporting Information files or to a stable, public repository and provide us with the relevant URLs, DOIs, or accession numbers. Please see http://www.bmj.com/content/340/bmj.c181.long for guidelines on how to de-identify and prepare clinical data for publication. For a list of acceptable repositories, please see http://journals.plos.org/plosone/s/data-availability#loc-recommended-repositories.

Additional Editor Comments :

Dear Authors,

after careful revision of the article and considering the reviewers comments, my decision as academic editor is "Major Revision".

Both reviewers have recognized clear interest in the article and I think too that your article is in line with PLOSONE criteria. However, in the current form there are certain aspects that, for the time being, preclude the acceptance of the article.

I suggest to follow point by point all the suggestions proposed by the reviewers.

You have granted standard time for the review, but if you need extra time, please do not hesitate in asking for it.

Thank you for your contribution to PLOSONE.

Dr. Oreste Gualillo, PharmD, PhD

PLOSONE Academic Editor.

Reviewers' comments:

Reviewer's Responses to Questions

**Comments to the Author**

1. Is the manuscript technically sound, and do the data support the conclusions?

Reviewer #1: Yes

Reviewer #2: Partly

2. Has the statistical analysis been performed appropriately and rigorously? 

Reviewer #1: No

Reviewer #2: Yes

3. Have the authors made all data underlying the findings in their manuscript fully available?

Reviewer #1: Yes

Reviewer #2: Yes

4. Is the manuscript presented in an intelligible fashion and written in standard English?

Reviewer #1: No

Reviewer #2: No

5. Review Comments to the Author

Reviewer #1: In this study, Rankothgedera S. et al aimed to identify potential serum immunologic diagnostic markers of primary and secondary knee osteoarthritis (pKOA and sKOA, respectively). Their results suggest that there might be distinct panels of markers for pKOA and sKOA that, if confirmed, may be useful for the development of therapeutic strategies for these two types of KOA. The study is well designed, the methodology is appropriate and the conclusions are supported by the results obtained.

However, there are some major issues that should be resolved prior to the acceptance of the manuscript:

1) The authors should define the abbreviations upon their first appearance in the text, and always use the same abbreviation (e.g. when talking about primary and secondary knee osteoarthritis, sometimes they refer to it as pKOA or sKOA, sometimes as primary/secondary OA). Please unify.

2) Additionally, after an abbreviation is defined for the first time, they should use the abbreviation thenceforth (e.g. RA, SLE).

3) In the Statistical Analyses sub-section the authors should detail which were the confounding factors included in the analysis.

4) In the tables the authors should define the abbreviations used in the legends, below them.

5) Figure 1 A and B: The type of graph chosen by the authors is not correct for the data/results they want to show. They should use a box plot or scatter plot, for example. But they should not add connecting lines between the values obtained in each study group for each marker.

6) In the Statistical Analyses sub-section the authors mention that the level of statistical significance was set at p<0.05, meaning that a p-value of 0.05 is not considered as statistically significant. However, some of the results reported in this manuscript as statistically significant exhibit a p-value = 0.05. Please check and correct.

7) The number of individuals with sKOA is relatively low. Accordingly, this should also be included as a potential limitation of the study, since the low sample size of this group may affect the results obtained (and the conclusions drawn from them).

In addition, there are some minor issues that should also be addressed:

1) Line 164: Some punctuation marks are missing in the caption of Figure 1. Please check.

2) Line 176: The correct phrase is ‘on par’, not ‘in par’.

3) Line 186: Please, cite Figure 2 at the end of the paragraph.

4) Line 242: The authors should replace ‘Females’ by ‘Female sex’.

5) Line 247: The term ‘that’ is repeated.

Reviewer #2: In the manuscript titled “A potential diagnostic serum immunological marker panel to differentiate between primary and secondary knee osteoarthritis” the authors identified a serum biomarker panels able to differentiate primary from secondary knee osteoarthritis (KOA).

The paper includes an adequate statistical analysis and provides experimental findings that allow to distinguish primary OA from secondary disease. However, there are some aspects, that should be considered to improve the overall completeness and quality of the paper.

1. It is not clear to this referee how the study was designed ; specifically, it is no specified timing of blood withdraw from KOA patients. Also, disease stage was not indicated. Please, consider these aspects also for SLE subjects. This are very important issues in order to determine the “predictive” role of the panel proposed.

2. In this context, the authors should provide a schematic diagram representing the study design. The authors investigated the of Th1/Th2 ratio on KOA and analyzed TNF�:IL-10 ratio as readout of Th1/Th2 balance. However, other cytokines contribute to this balance (e.g, IFN-a, IL-2, IL-12 Vs IL-4, IL-5); some of these additional cytokines should be measured to better define Th1/Th2 balance.

Throughout the manuscript the data are not well presented. As relevant examples, figure 1 (A & B) and figure 2 are not clear. In this context, box plot with dots should be used to improve the message from these findings.

6. PLOS authors have the option to publish the peer review history of their article (what does this mean?). If published, this will include your full peer review and any attached files.

Reviewer #1: No

Reviewer #2: No

---

## [Author Response · Author response to Decision Letter 0]

23 Aug 2021

Responses to reviewers’ comments:

PONE-D-21-12809: A potential diagnostic serum immunological marker panel to differentiate between Primary and Secondary Knee Osteoarthritis

As coauthors of this manuscript, we deeply appreciate the constructive comments and suggestions made by the reviewers; we have attempted to address all queries raised by the two reviewers. We hope our manuscript now meets the caliber of the journal and will be suitable to be published in PLOS ONE.

Reviewer #1: In this study, Rankothgedera S. et al aimed to identify potential serum immunologic diagnostic markers of primary and secondary knee osteoarthritis (pKOA and sKOA, respectively). Their results suggest that there might be distinct panels of markers for pKOA and sKOA that, if confirmed, may be useful for the development of therapeutic strategies for these two types of KOA. The study is well designed, the methodology is appropriate and the conclusions are supported by the results obtained.

However, there are some major issues that should be resolved prior to the acceptance of the manuscript:

1) The authors should define the abbreviations upon their first appearance in the text, and always use the same abbreviation (e.g. when talking about primary and secondary knee osteoarthritis, sometimes they refer to it as pKOA or sKOA, sometimes as primary/secondary OA). Please unify.

The suggested changes were made in the revised manuscript. 

2) Additionally, after an abbreviation is defined for the first time, they should use the abbreviation thenceforth (e.g. RA, SLE).

The original manuscript was revised based on the reviewer comment. 

3) In the Statistical Analyses sub-section the authors should detail which were the confounding factors included in the analysis.

Confounding factors used in the analysis were included in the Statistical Analyses sub-section of the revised manuscript (lines 150-151).

4) In the tables the authors should define the abbreviations used in the legends, below them.

This shortcoming in Tables 1 to 4 was corrected in the revised manuscript.

5) Figure 1 A and B: The type of graph chosen by the authors is not correct for the data/results they want to show. They should use a box plot or scatter plot, for example. But they should not add connecting lines between the values obtained in each study group for each marker.

Taking into consideration the reviewer’s comment, Figures 1 and 2 (Figs 2 and 3 in the revised manuscript) are presented as box plots (with dots), in the revised manuscript.

6) In the Statistical Analyses sub-section the authors mention that the level of statistical significance was set at p<0.05, meaning that a p-value of 0.05 is not considered as statistically significant. However, some of the results reported in this manuscript as statistically significant exhibit a p-value = 0.05. Please check and correct.

In the Statistical Analyses sub – section, the last sentence in the revised manuscript reads as “The level of statistical significance was set at p≤0.05” (line 154).

7) The number of individuals with sKOA is relatively low. Accordingly, this should also be included as a potential limitation of the study, since the low sample size of this group may affect the results obtained (and the conclusions drawn from them).

The sentence, “Nonetheless, the low sample size of the sKOA group may present as a potential limitation of this study, which may perhaps have ramifications (lines 289-291)” was inserted. 

In addition, there are some minor issues that should also be addressed: 

1) Line 164: Some punctuation marks are missing in the caption of Figure 1. Please check.

2) Line 176: The correct phrase is ‘on par’, not ‘in par’.

3) Line 186: Please, cite Figure 2 at the end of the paragraph.

4) Line 242: The authors should replace ‘Females’ by ‘Female sex’.

5) Line 247: The term ‘that’ is repeated.

The aforementioned minor corrections were rectified in the revised manuscript.

Reviewer #2: In the manuscript titled “A potential diagnostic serum immunological marker panel to differentiate between primary and secondary knee osteoarthritis” the authors identified a serum biomarker panels able to differentiate primary from secondary knee osteoarthritis (KOA).

The paper includes an adequate statistical analysis and provides experimental findings that allow to distinguish primary OA from secondary disease. However, there are some aspects that should be considered to improve the overall completeness and quality of the paper.

1. It is not clear to this referee how the study was designed; specifically, it is no specified timing of blood withdrawal from KOA patients. Also, disease stage was not indicated. Please, consider these aspects also for SLE subjects. These are very important issues in order to determine the “predictive” role of the panel proposed. In this context, the authors should provide a schematic diagram representing the study design.

The following revisions were incorporated to the revised manuscript:

Under Materials and Methods, sub section ‘Study Participants’

• The case control study design (scheme) was incorporated as Figure 1 (lines 102-103),

• Both pKOA (line 112) and sKOA (line 116) patients recruited for the study were at Stage 3 of the disorders. 

• All SLE patients recruited as disease controls for the study, specifically manifested lupus arthritis (line 109-110).

. 

2. The authors investigated the Th1/Th2 ratio on KOA and analyzed TNF�:IL-10 ratio as readout of Th1/Th2 balance. However, other cytokines contribute to this balance (e.g, IFN-a, IL-2, IL-12 Vs IL-4, IL-5); some of these additional cytokines should be measured to better define Th1/Th2 balance. 

While completely agreeing with the reviewer on this viewpoint, lack of funds restricted us to assay selected serum cytokines. Hence, the sentences “However, the assay of additional serum cytokines such as IFN-γ, IL-2, IL-12 Vs IL-4, IL-5 would have explicitly defined the Th1:Th2 balance among the study groups. This was not pragmatic due to restricted funds” were inserted (lines 197-199). 

3.Throughout the manuscript the data are not well presented. As relevant examples, figure 1 (A & B) and figure 2 are not clear. In this context, box plot with dots should be used to improve the message from these findings.

Taking into consideration the reviewer’s comment, Figures 1 and 2 (Figs 2 and 3 in the revised manuscript) are presented as box plots (with dots), in the revised manuscript.

---

## [Decision Letter · Decision Letter 1]

3 Sep 2021

A potential diagnostic serum immunological marker panel to differentiate between Primary and Secondary Knee Osteoarthritis

PONE-D-21-12809R1

Dear Dr. Udagama,

We’re pleased to inform you that your manuscript has been judged scientifically suitable for publication and will be formally accepted for publication once it meets all outstanding technical requirements.

Kind regards,

Oreste Gualillo, PharmD, PhD

Academic Editor

PLOS ONE

Additional Editor Comments (optional):

No additional comments.

Reviewers' comments:

Reviewer's Responses to Questions

**Comments to the Author**

1. If the authors have adequately addressed your comments raised in a previous round of review and you feel that this manuscript is now acceptable for publication, you may indicate that here to bypass the “Comments to the Author” section, enter your conflict of interest statement in the “Confidential to Editor” section, and submit your "Accept" recommendation.

Reviewer #2: All comments have been addressed

2. Is the manuscript technically sound, and do the data support the conclusions?

Reviewer #2: Yes

3. Has the statistical analysis been performed appropriately and rigorously? 

Reviewer #2: Yes

4. Have the authors made all data underlying the findings in their manuscript fully available?

Reviewer #2: Yes

5. Is the manuscript presented in an intelligible fashion and written in standard English?

Reviewer #2: Yes

6. Review Comments to the Author

Reviewer #2: The authors addresse most of the prevoius concerns. I suggest the to remobe the sentence "this was not pragrmatic due the restrict fund". Add the absence of Th1/Th2 characterization as one of the limitations of the study.

7. PLOS authors have the option to publish the peer review history of their article (what does this mean?). If published, this will include your full peer review and any attached files.

Reviewer #2: No

---

## [Editor Report · Acceptance letter]

9 Sep 2021

PONE-D-21-12809R1 

A potential diagnostic serum immunological marker panel to differentiate between Primary and Secondary Knee Osteoarthritis 

Dear Dr. Udagama:

I'm pleased to inform you that your manuscript has been deemed suitable for publication in PLOS ONE. Congratulations! Your manuscript is now with our production department. 

Kind regards, 

on behalf of

Dr Oreste Gualillo 

Academic Editor

PLOS ONE